# Comparison between Classical- and Rotational-Mechanical-Chair-Assisted Maneuvers in a Population of Patients with Benign Paroxysmal Positional Vertigo

**DOI:** 10.3390/jcm13133863

**Published:** 2024-06-30

**Authors:** Marta Chaure-Cordero, Maria Garrote-Garrote, Jonathan Esteban-Sánchez, Paula Morales-Chacchi, Marina Del Valle-Díaz, Eduardo Martin-Sanz

**Affiliations:** 1Department of Otolaryngology, Getafe University Hospital, Carretera Toledo km 12,500, 28905 Madrid, Spain; m.garrotega@gmail.com (M.G.-G.); jonathanestebansanchez@gmail.com (J.E.-S.); pmpaulamorales@gmail.com (P.M.-C.); marina.delvalle.diaz@gmail.com (M.D.V.-D.); emartinsanz@gmail.com (E.M.-S.); 2Department of Medicine, School of Biomedical Sciences and Health, Universidad Europea de Madrid, 28670 Villaviciosa de Odón, Spain

**Keywords:** benign paroxysmal positional vertigo, vertigo, TRV chair, mechanical rotational chair, repositioning maneuvers

## Abstract

**Introduction**: Benign paroxysmal positional vertigo (BPPV) stands as the most common cause of peripheral vertigo. Its treatment with repositioning maneuvers on an examination table is highly effective. However, patients with back or neck problems, paraplegia, or other conditions face challenges with these maneuvers, potentially experiencing longer healing times and creating additional difficulties for physicians diagnosing and treating BPPV in everyday practice. The emergence of mechanical rotational chairs (MRCs) offers a more convenient alternative for performing these maneuvers. **Objectives:** The primary objective was to compare the effectiveness of maneuvers on the examination table with those on MRCs in BPPV patients diagnosed in the emergency room and randomly classified into one of the treatment options. The secondary objectives included a comparison of patient quality of life during BPPV episodes and after their resolution and an analysis of recurrences and associated risk factors. **Methods:** This was a cohort study on sixty-three patients diagnosed with BPPV in the emergency department. Patients were classified into two cohorts depending on diagnostic and treatment maneuvers (MRC or conventional repositioning maneuvers (CRMs)) and received weekly follow-ups until positioning maneuvers became negative. Subsequent follow-ups were conducted at 1 month, 3 months, and 6 months after the resolution of vertigo. Patients were classified into two groups based on their assigned treatment method. **Results:** Thirty-one patients were treated with CRMs and 32 with TRV. Mean age was 62.29 ± 17.67 years and the most affected canal was the PSC (96.8%). The mean number of required maneuvers was two, while 55.56% only required one maneuver until resolution. Recurrence was present in 26.98% of the patients during the 6-month follow-up. Comparing both cohorts, there were no statistically significant differences between treatments (TRV vs. CRM) regarding the number of maneuvers, number of recurrences, and days until remission of BPPV. Dizziness Handicap Inventory and Visual Analogue Scale values decreased considerably after BPPV resolution, with no statistically significant differences between the groups. Age was identified as a covariable in the number of maneuvers and days until BPPV resolution, showing that an increase in age implies a greater need of maneuvers. **Conclusions:** There was no difference between the means of treatment for BPPV in our population ot There was no difference between the groups of treatments for BPPV in our population. The quality of life of patients improved six months after the resolution of BPPV, regardless of the treatment applied.

## 1. Introduction

Benign paroxysmal positional vertigo (BPPV) is the most common cause of peripheral vestibular disorders. It is caused by the displacement of otoconia from the utricle in one or more semicircular canals (SCC), and it is characterized by brief episodes (half to one minute) of rotatory vertigo triggered by specific head movements [1]. Otoconia can be floating in the endolymphatic space of the SCC (canalolithiasis) or be attached to the cupula within the SCC (cupulolithiasis) [2,3].

In the literature, a female-to-male ratio of 2:1 is described, with involvement of the right posterior semicircular canal (PSC) in approximately 72–80% of cases. The average age onset ranges between 49 and 57 years old [4,5]. Nevertheless, the involvement of other semicircular canals along with other rare variants of BPPV often lead to misdiagnosis.

Although the disease is benign, in addition to the vertigo and vegetative symptoms, BPPV can be very debilitating, producing psychological and physical impairments that decrease the quality of life. It also increases the risk of falls and fear of falling, especially in elderly patients [6,7]. However, once the episode is resolved, an improvement in these symptoms is observed, although some patients may still experience chronic subjective symptoms [8].

Up to 50% of BPPV cases resolve spontaneously in 2 to 12 weeks [5,9,10], and between 90 and 98% resolve after one or two repositioning maneuvers [11]. However, a specific group of patients with high risk of BPPV needs more treatment sessions and has a higher risk of recurrence. This group includes those with a history of head trauma, reduced head mobility, and the geriatric population [12].

The management of BPPV involves canalith repositioning maneuvers, such as the Epley or Semont maneuver [13,14], whose aim is to relocate the otoconia from the PSC back into the utricle. Maneuvers are typically performed manually on the examination table. However, it is important to note that the success of these maneuvers relies on their correct execution by the examiner, inter-examiner variability, and specific factors of the subject that may hinder the satisfactory completion of the maneuver and can even increase the risk of multicanal involvement if inappropriately performed. Similarly, maneuvers performed by patients at home may result in a lower cure rate. Patients with conditions such as obesity, cervical arthritis, paraplegia, or other mobility disorders face challenges in carrying out these maneuvers [15]. Considering these limitations, mechanical rotational chairs (MRC) have been developed for the diagnosis and treatment of BPPV. These chairs allow for 360-degree movements to perform all the described BPPV maneuvers. They enable specific angles and inclinations for each SCC, as well as more abrupt movements with greater acceleration force, without altering the position of the patient’s head or requiring neck hyperflexion. The patient remains seated in the same position throughout the procedure [16].

Currently, there are four different chairs available: the Rotundum repositioning chair, the Thomas Richard-Vitton repositional chair (TRV) [16], the Epley Omniax System (EO), and the Automated Mechanical Repositioning Treatment [17].

So far, studies conducted with this tool have shown that MRCs are superior in the treatment of BPPV patients with multicanal or bilateral BPPV, cupulolithiasis, or refractory vertigo [18]. However, it is important to note that most of these studies are retrospective, non-randomized, and lack a systematic and protocolized follow-up [19,20,21]. As a result, there is a lack of clinical evidence regarding the true superiority or benefits of MRCs compared to traditional maneuvers in a population that accurately represents the everyday reality of an emergency and ENT unit in any healthcare facility.

Taking into consideration the information above, we developed a cohort study that compares the efficacy of both means of treatment of BPPV available in our facility.

## 2. Materials and Methods

### 2.1. Patient Selection

Sixty-three patients were recruited for this cohort study from March 2023 to February 2024 by members of the Department of Otorhinolaryngology in Getafe University Hospital. Patients with a recent history of labyrinthine disorders (including vestibular neuritis, acute sensorineural hearing loss, and Ménière’s disease) were excluded from this study to avoid introducing confounding factors related to symptoms and quality of life. This decision was made because diagnosing and treating BPPV in these cases is more challenging, and the presence of another vestibular disorder can act as a confounding factor. Other pathologies, such as vestibular migraine or any other disorders of the central nervous system were excluded with a complete neurological assessment. A cerebral MRI was performed in cases of doubt regarding neurological etiology. Additionally, pregnant women and patients under 18 years old were excluded.

Patients were divided into two cohorts depending on whether the diagnostic and treatment maneuvers were undertaken using a TRV MRC (TRV chair, Interacoustics, Middelfart, Denmark) or conventional repositioning maneuvers (CRMs) on the examination table.

This project was approved by our institutional review board (CEIm22/58). Informed consent was obtained from every patient.

### 2.2. Patient Assessment

Patients were admitted to the emergency unit and were diagnosed with BPPV according to the criteria formulated by the Committee for Classification of Vestibular Disorders of the Bárány Society (clinical presentation: recurrent episodes of vertigo triggered by changes in head position; duration: seconds to several minutes; frequency: the episodes occur frequently, often several times a day; trigger: the episodes are triggered by specific head movements, such as rolling over in bed, getting up from a lying down position, or looking up or down; nystagmus: the patient exhibits nystagmus (abnormal eye movements) during the episodes; no other causes: the patient does not have any other underlying conditions that could cause vertigo, such as vestibular migraine, Ménière’s disease, or labyrinthitis) [22]. Patients were always treated with CRM on the first visit as they were seen in the emergency department.

Upon enrollment in this study, each patient underwent comprehensive evaluation conducted by the same team on every occasion. This evaluation encompassed a thorough review of clinical history and neurotological physical examination. Eye movement was recorded using goggles equipped with an infrared camera (Visual Eyes™ 505, Interacoustics, Denmark).

Patients with prior episodes of BPPV had not undergone treatment with any mechanical rotational chair, instead receiving conventional maneuvers, and the most recent episode had occurred at least 6 months before the onset of the new episode.

Every patient underwent weekly examinations, including specific diagnostic maneuvers for each semicircular canal (Dix–Hallpike, McClure, and the Head Hanging position) and specific repositioning maneuvers tailored to the affected semicircular canal until there was no nystagmus seen during positioning maneuvers and no vertigo symptoms triggered. Subsequently, they received follow-up appointments at one month, three months, and six months after being treated.

During the follow-up, patients underwent complete neurotological evaluation and repositioning maneuvers, if needed, depending on the cohort they were included in. They filled in the Dizziness Handicap Inventory (DHI) and Visual Analogue Scale (VAS) on every visit and the Short Falls Efficacy Scale International (Short FES-I) on the second visit and a month after cure of BPPV. Figure 1 shows the flowchart followed in every patient and the specific maneuver applied depending on the mean of treatment.

In case of recurrence, we evaluated the patient again on a weekly basis until the negativization of the diagnostic maneuvers.

Appendix A depict the execution of maneuvers in each group.

### 2.3. Handicap Measurements

We used the DHI questionnaire [23,24], while vertigo severity was assessed using the VAS. DHI was developed in 1990 and translated into and validated in Spanish in 2000. It consists of 25 questions assessing the physical, emotional, and functional impact of vertigo on daily life, with scores ranging from 0 to 100, with higher scores indicating greater handicap. The DHI is simple, reliable, and allows for comprehensive evaluation of the patient’s condition.

VAS scores were obtained by asking the patient to rate the severity of their vertigo from 0 to 10.

Additionally, a risk of falls scale, the Short FES-I was filled out [25]. Short FES-I evaluates how concerned the patient is about the possibility of falling while performing seven activities, rating each one from “not concerned at all”, “somewhat concerned”, and “fairly concerned” to “very concerned”, and has a total score from 7 to 28.

### 2.4. Statistical Analysis

All data were processed using the Statistical Analysis Statistical Package for the Social Sciences for Windows (SPSS version 25.0; IBM Corp, Armonk, NY, USA).

The normality of the study population’s distribution was assessed using the Kolmogorov–Smirnov test.

Values were described as the mean ± standard deviation. For the evolution of continuous quantitative variables, the ANOVA test was used. For qualitative variables, Chi2 test was performed, and the *t-*test was employed to compare quantitative variables between groups. A study of bivariate correlation was performed to analyze the impact of age and body mass index (BMI) on the number of recurrences, days until remission of BPPV, and number of maneuvers. Statistical significance for all tests was considered at a *p-*value < 0.05.

## 3. Results

Sixty-three subjects were included in this study and completed the treatment and follow-up. A total of 32 of the 63 patients (50.7%) were randomized to treatment with the TRV chair, and 31 of the 63 subjects (49.3%) were randomized to traditional treatment on an examination bed.

Eighteen of the patients (28.6%) were male and forty-five (71.4%) were female. Mean age was 62.29 ± 17.67 years in the global population. The most affected canal was the PSC, with a total of 96.8% of the population. The left ear was affected in 54% of the population and the right ear in 42.9%. In total, 14.29% (9/63) of patients had a bilateral affection. These findings are shown in Table 1. No statistical differences were found in the distribution of population characteristics between groups based on gender, age, affected side and semicircular canal, or body mass index, making the groups statistically comparable.

Mean follow-up time of the population was 196.97 ± 23.87 (6.5 months).

### 3.1. Risk Factors Distribution

Sixteen out of thirty-two (50%) patients treated using TRV had previously suffered at least one episode of BPPV, versus nineteen out of thirty-one (61.29%) in the CRM group. Table 2 summarizes the frequencies of all the studied risk factors in each group. There were non-significant differences between the groups when comparing the possible risk factors for BPPV.

### 3.2. Number of Treatments and Days until Resolution of BPPV

The TRV group required a mean of 2.34 ± 0.56 maneuvers to achieve remission of BPPV, whereas the CRM group needed 1.71 ± 0.21 (Figure 2). There was no statistically significant difference in the number of treatments required between the two groups in our population (*p* = 0.120). There were five patients in the TRV group and two in the manual treatment group requiring five or more maneuvers, of which four had bilateral BPPV. Table 3 shows the number of patients that needed one maneuver, between two and four and more than five in each group. 

The mean number of days until achieving a negative maneuver in the global population was 15.08 (±12.14); specifically, in the TRV group, it was 17.75 (±2.53) days and in the manual treatment group, 12.32 (±1.57) (Figure 3). No significant differences were found between the groups (*p* = 0.076).

However, the Pearson correlation test showed a significant correlation between age and days until cure (*r* = 0.246; *p* = 0.026) and number of maneuvers applied (*r* = 0.296; *p* = 0.009) (Figure 4). Despite being significant, these results indicate a weak correlation. On the ANOVA test, it was also shown that age is a covariable in the number of maneuvers (*p* = 0.026) and days until remission (*p* = 0.024).

### 3.3. Quality of Life Evaluation: DHI, VAS and Sort FES-I Scores

Table 4 summarizes quality of life test scores in each group.

Quality of life was reflected in significantly lower DHI total scores and VAS scores in follow-up visits after successful treatment (*p* < 0.001) (Figure 5). Nevertheless, the comparison of the scores between both treatment groups did not show statistically significant differences in either first visit scores or post-treatment scores (Table 4).

### 3.4. Recurrence

Seventeen out of sixty-three (26.98%) subjects suffered a recurrence within the six-month follow-up. The TRV group had a 25% recurrence (8/32) and the CRM group had a recurrence rate of 29.03% (9/31), showing no statistically significant differences (*p* = 0.718). Of all the recurrences, only four patients experienced a canal shift in both groups. All cases shifted from HSC canalolithiasis to PSC; three subjects were in the TRV group and one patient in the CRM group.

There was a moderate positive correlation between BMI and the number of recurrences (*r* = 0.514; *p* = 0.017).

## 4. Discussion

The present study represents one of the first prospective randomized studies which compare the treatment of common BPPV patients with mechanical chairs versus manual repositioning maneuvers. It is also one of the few prospective and randomized studies with a long-term follow-up of patients treated by the same team every time and referred directly to the ENT department right after their first visit in the emergency room. Many studies related to this topic do not have a control group [19]; select only complex patients with multicanal BPPV [26]; or compare the efficacy of different mechanical rotational chairs with manual maneuvers [21].

In the same way, many other studies are retrospective [17,26]; do not show a long-term follow-up [27]; or the period between visits while the patients are affected with BPPV is too long [19].

In our study, both cohorts were comparable as they did not differ with respect to demographic or clinical baseline characteristics. The average age of our population was slightly higher than in some published series [5], whereas other studies show similar mean age [17]. The female-to-male ratio was higher, as we found more women than men, as seen in the majority of neurotological disorders [4,28]. According to which canal was affected and which side, we found differences between studies, some show a similar ratio of left/right affection [28], whereas others show a higher affection of the right labyrinth [29]; in addition, we found a significantly higher incidence of bilateral BPPV, which can be challenging to treat [4]. These differences between studies can be explained by the challenges many clinicians face when diagnosing BPPV, as there are no clear objective tests for it, and the diagnosis depends on the clinical experience of the physician and on the collaboration of the patient [30]. These demographic differences may also affect treatment efficacy but are mostly dependent on the type of hospital and neurotologic unit providing the treatment.

### 4.1. Treatment Efficacy

Our results, showing a successful treatment rate, with 55.56% of our patients being cured after the first visit, are comparable to other studies, where rates range from 34.2% to 87% [13,28,31,32]. These findings align with the literature because every patient in our study received manual treatment during their first visit. This is due to patient recruitment being conducted through the emergency department, where only an examination table is available, after diagnosis and initial treatment. This represents a common clinical scenario, as BPPV patients are treated with manual maneuvers in most cases. However, this is a limitation of our study, and in future studies, it would be beneficial to compare two groups of patients with acute vertigo, each receiving the specific maneuver for their respective treatment group (TRV or CRM) during the first visit.

This wide range of success rates found in the literature could be explained by the variety of protocols followed in each study, the fact that not all patients co-operate equally in the execution of the maneuver [15], the differences in the follow-up periods, or the experience of the treating clinician, as some reports were run by general practitioners [33].

Regarding treatment, both means of treatment were equal in terms of the number of maneuvers and days to achieve remission of BPPV, as no statistically significant differences were found. Research on this topic yields very variable results. Tan et al. conducted a prospective randomized study on PSC-BPPV that showed statistically significant differences between groups in mean numbers of maneuvers on the first follow-up visit. However, there were no differences in long-term follow-up [20]. On the other hand, one study that retrospectively evaluated the treatment of multicanal BPPV with TRV or manual treatment showed no statistically significant differences [26]. There is only one other study that resembled ours the most in terms of design, even though it only included short-term follow-up. The study did not show differences between groups in terms of the number of maneuvers [27]. All previously published studies only evaluated the number of treatments but did not show the time interval between visits/treatments, and we consider this an important fact, as long periods between follow-ups can lead to biases due to spontaneous resolution of BPPV [34,35,36]. This was considered when designing our study protocol to minimize the possibility of spontaneous resolution.

The same studies that did not find differences in the number of maneuvers, especially the retrospective study by Baydan-Aran et al. [26] and the one by Luryi et al. [36], showed a similar mean number of treatments in each group, which matches our study.

A stratification of our population into age groups shows that the older the subjects, the more maneuvers and days until resolution they need. Nevertheless, we can see a tendency of subjects in the TRV group to achieve resolution faster than the manual treatment group (Figure 4). This can be explained by the fact that the elderly have a more restricted mobility of the neck and of the whole body at the time of performing the repositioning maneuvers [15] and might represent a selected at-risk population who could benefit from treatment using a TRV chair.

### 4.2. Quality of Life

Questionnaires answered by our population show a statistically significant improvement in scores, with no significant differences between groups. This is consistent with other research on this topic analyzing BPPV groups treated using mechanical rotational chairs [27,37,38] or manual procedures [6,8].

### 4.3. Recurrence

Up to 50% of patients experience a recurrence of BPPV within the next 10 years, with 80% of these occurring in the first year [38]. In many studies evaluating recurrences in BPPV treated with MRC, it is shown that recurrences happen mostly in the first six months and that these comprise 25% of recurrences in this population, which matches our findings (26.98%) [17]. However, many of these studies do not have a long-term or uniform and consistent follow-up for each subject in the population. Thus, a longer follow-up period should be considered in our study to evaluate recurrence rates beyond the six-month period.

## 5. Conclusions

In conclusion, the TRV chair proves to be a safe tool for both diagnosing and treating BPPV, without an increased risk of recurrences. However, it is not superior to manual treatment in conventional BPPV patients referred to our unit after a first treatment on the examination table in the emergency room. Nevertheless, TRV seems to be more effective in elderly patients and more convenient in populations with mobility difficulties.

Quality of life improves considerably after the resolution of BPPV no matter the means of treatment. Further studies need to be conducted to analyze the long-term follow-up of these two groups of patients prospectively. It would also be appropriate to have a bigger population to study bigger subgroups of patients (age, type of BPPV, risk factors…).

## Figures and Tables

**Figure 1 jcm-13-03863-f001:**
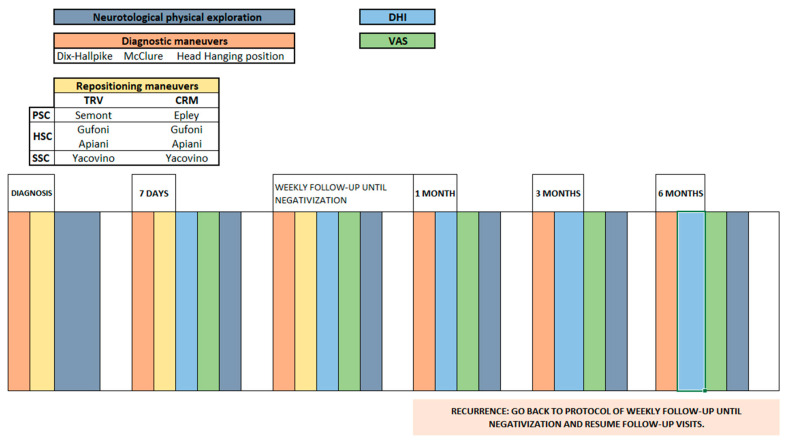
Flowchart of study protocol. TRV: Thomas Richard-Vitton, CRM: conventional repositioning maneuvers; PSC: posterior semicircular canal, HSC: horizontal semicircular canal, SSC: superior semicircular canal; VAS: Visual Analogue Scale; DHI: Dizziness Handicap Inventory.

**Figure 2 jcm-13-03863-f002:**
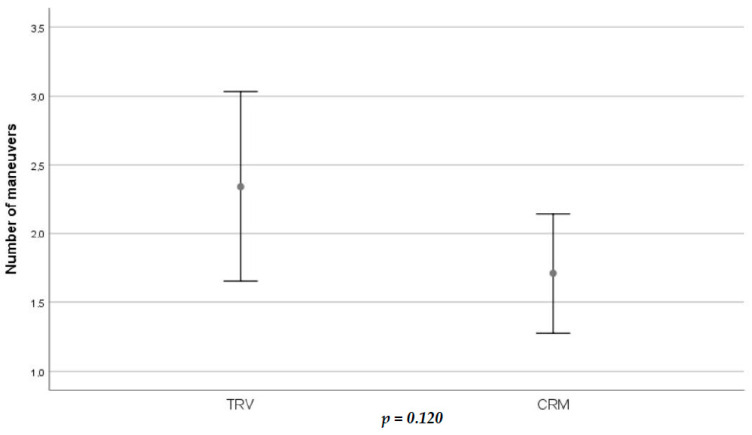
Error bar showing the number of maneuvers needed to get a successful treatment in each group. *T* student test with statistical significance level defined as *p-*value < 0.05.

**Figure 3 jcm-13-03863-f003:**
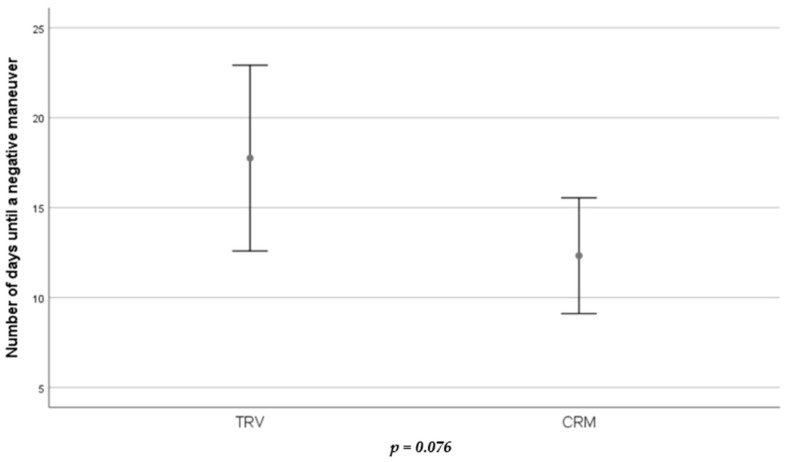
Error bar showing the number of days needed to get a successful treatment in each group. *T* student test with statistical significance level defined as *p-*value < 0.05.

**Figure 4 jcm-13-03863-f004:**
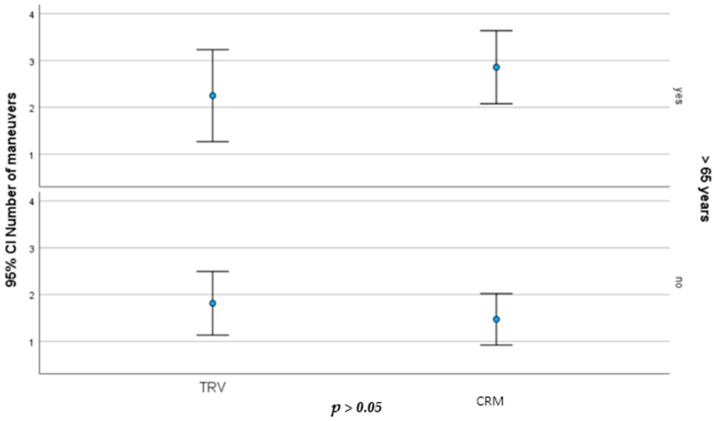
Error bar showing less maneuvers in the TRV group in patients of more than 65 years old.

**Figure 5 jcm-13-03863-f005:**
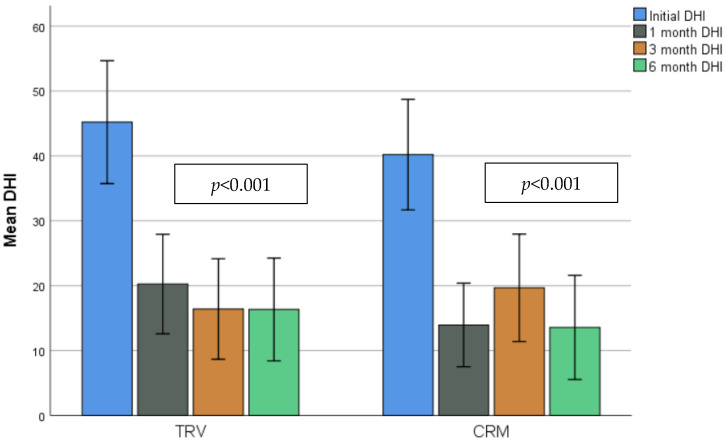
Bar chart showing evolution of DHI total scores in initial and follow-up visits. ANOVA test with a statistical significance defined as *p-*value < 0.05.

**Table 1 jcm-13-03863-t001:** Population characteristics.

	TRV ^1^ (*n* = 32)	CRM ^2^ (*n* = 31)	TOTAL	*p*-Value *
Sex, *n* (%)				0.11
- Male	14 (43.75%)	4 (12.9%)	18 (28.57%)	
- Female	18 (56.25%)	27 (87.1%)	45 (71.43%)	
Age (years), mean (SD ^3^)	63.28 (17.32)	61.26 (18.25)	62.29 (17.67)	0.654
BMI ^4^ (Kg/m^2^), mean (SD)	26.15 (4.21)	19.73 (18.94)	28.55 (13.73)	0.171
Affected canal, *n* (%)				1
- PSC	31 (49.21%)	30 (47.61%)	61 (96.82%)	
- LSC	1 (1.58%)	1 (1.58%)	2 (3.16%)	
Side *n* (%)				0.118
- Left	17 (26.99%)	10 (15.87%)	27 (42.86%)	
- Right	15 (23.81%)	19 (30.16%)	34 (53.97%)	
- Bilateral	0 (0%)	2 (3.17%)	(3.17%)	

^1.^ Thomas Richard-Vitton (TRV). ^2.^ Conventional repositioning maneuvers (CRMs). ^3.^ Standard deviation (SD). ^4.^ Body mass index (BMI). * Chi-squared test for qualitative variables (gender, affected canal and side); Student’s *t*-test for quantitative variables (age and BMI). Statistical significance was defined as *p-*value < 0.05.

**Table 2 jcm-13-03863-t002:** Risk factor distribution.

	TRV ^1^ (*n* = 32)	CRM ^2^ (*n* = 31)	TOTAL	*p*-Value ^*^
History of BPPV, *n* (%)	16 (50%)	19 (61.29%)	35 (55.56%)	0.450
Low vitamin D	9 (28.13%)	10 (32.26%)	19 (30.16%)	0.788
Head trauma	3 (9.37%)	2 (6.45%)	5 (7.94%)	1
Falls	4 (12.5%)	2 (6.45%)	6 (9.52%)	0.672
Dyslipidemia	4 (12.5%)	10 (32.26%)	14 (22.22%)	0.075
Osteoporosis	2 (6.25%)	7 (22.58%)	9 (14.29%)	0.082
Recent surgery	1 (3.13)	2 (6.45%)	3 (4.76%)	0.613

^1.^ Thomas Richard-Vitton (TRV). ^2.^ Conventional repositioning maneuvers (CRMs). * Chi-squared test. Statistical significance was defined as *p-*value < 0.05.

**Table 3 jcm-13-03863-t003:** Number of maneuvers.

	1	2–4	≥5
TRV ^1^ group (*n* = 32), *n* (%)	15 (46.87%)	12 (37.5%)	5 (15.63%)
CRM ^2^ (*n* = 31), *n* (%)	20 (64.52%)	9 (29.03%)	2 (6.45%)
TOTAL	35 (55.56%)	21 (33.33%)	(11.11%)

^1.^ Thomas Richard-Vitton (TRV). ^2.^ Conventional repositioning maneuvers (CRMs).

**Table 4 jcm-13-03863-t004:** Quality of life scores. Student’s *t-*test with statistical significance level defined as *p-*value < 0.05.

		First Visit	*p*-Value	One Month	*p*-Value	Three Months	*p*-Value	Six Months	*p*-Value
Short FES-I ^1^	TRV (*n* = 32), mean (SD)	9.22 (2.68)	0.656	7.72 (1.25)	0.317				
	CRM (*n* = 31), mean (SD)	9.58 (3.64)		7.45 (0.81)					
VAS ^2^	TRV (*n* = 32), mean (SD)	5.42 (2.38)	0.292	2.16 (2)	0.265	1.61 (1.5)	0.364	1.61 (1.78)	0.273
	CRM (*n* = 31), mean (SD)	4.1 (1.84)		1.63 (1.71)		1.23 (1.82)		1.08 (1.95)	
DHI (mean total score) ^3^	TRV (*n* = 32), mean (SD)	45.19 (26.28)	0.427	20.25 (21.26)	0.203	16.4 (20.79)	0.559	16.33 (21.27)	0.618
	CRM (*n* = 31), mean (SD)	40.19 (23.22)		13.94 (17.57)		19.67 (22.2)		13.57 (21.53)	

^1^ Short Falls Efficacy Scale International (Short FES-I) ^2^ Visual Analogue Scale (VAS). ^3^ Dizziness Handicap Inventory (DHI).

## Data Availability

The data presented in this study are available on request from the corresponding author due to the privacy of the population involved in this study.

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
