# Peer review of "Comparison between Classical- and Rotational-Mechanical-Chair-Assisted Maneuvers in a Population of Patients with Benign Paroxysmal Positional Vertigo"

_jcm, 2024, doi:10.3390/jcm13133863_

Round 1

Reviewer 1 Report

Comments and Suggestions for Authors

First, I would like to thank you for reading this manuscript. The present investigation deals with the comparison between conventional canalith repositioning manoeuvres and TRV rotational chair in treating BPPV. Since this is a relatively new method, this investigation is of interest. However, many issues must be corrected before this manuscript can be considered for publication.

Abstract

Lines 23-28. In addition to treating BPPV with specific manoeuvres, I recommend mentioning that in everyday practice, examiners can also face the mentioned difficulties when performing the diagnostic manoeuvres (e.g., the Dix-Hallpike) for BPPV. Mechanical-rotational-chairs are also essential considering this issue, as they can be used for treating and diagnosing BPPV. 

The abstract should concisely reflect the main study findings and methods; therefore, the details on the study population must be included.

Lines 41-42. It would be beneficial to define these ‘treatment groups’.

Lines 45-46. I recommend specifying the influence of age, i.e., older patients needed more manoeuvres for symptom improvement.

Introduction

Lines 54-55. It must be clarified that BPPV is the most common cause of peripheral vestibular disorders.

Line 64. The vertigo characteristics in BPPV would be beneficial to mention, i.e., vertigo lasting for half or one minute induced by head movements.

Lines 67-68. Regarding this statement, it must be mentioned that in cases when BPPV is not accurately treated, residual BPPV can even result in chronic symptoms.

Lines 77-79. Considering the effectiveness of conventional canalith repositioning manoeuvres, the limitations of ‘self-treatment’, i.e., when patients perform the repositioning manoeuvres at home, must also be mentioned. Furthermore, using conventional manoeuvres, when inappropriately performed, presents a higher risk for multicanal involvement. 

Materials and Methods

Line 105. Can the authors specify what they mean by excluding sudden sensorineural hearing loss? Does this mean that patients with acute vertigo accompanied by an acute sensorineural hearing loss were excluded? Otherwise, from my point of view, this statement does not make any sense since patients can have previous acute sensorineural hearing loss independently from a later manifested BPPV. On the other hand, it is known that BPPV can be secondary related to Ménière’s disease and vestibular neuritis. Although the diagnosis and treatment of these types are more challenging, they are practically the same as in primary cases. The exclusion of these cases must at least be explained to readers. Furthermore, the spelling of Ménière’s disease should be corrected.

Line 106. It is not mentioned how central nervous system disorders were excluded.  

Line 117. Since BPPV is still underdiagnosed, including its diagnostic criteria according to the Bárány Society would be beneficial.

Line 119. Instead of ‘anamnestic data’, case history would be a better term.

Line 122. ‘including all diagnostic maneuvers’- this statement is too general; the used manoeuvres according to canal involvement must be named.

Line 124. Since this is not an ENT-specific journal, ‘negativization’ would be beneficial to specify.

Regarding the study population, I missed mentioning the average symptom duration and whether this was statistically significant between the two study groups. The diagnosis and treatment of BPPV depend on symptom duration, especially in cases when patients were previously treated. Therefore, this information can significantly impact the results. 

Regarding the methods, it would be beneficial to use the following subheadings, including detailed information in each:

-       - Neurotological examinations (it is stated that a neurotological examination was performed; however, what this examination included needs to be clarified, i.e., only physical examination or objective testing?)

-       - CRMs (exact details depending on canal involvement)

-     - TRV (including manufacturing details) 

-     -  self-reported questionnaires (some more information on the DHI would be beneficial, e.g., its subscores, interpretation, and a statement about whether it was validated in Spanish)

Figure 1. Once again, neurotological exploration and diagnostic manoeuvres are too general; some details would even be beneficial included in this Figure.

Statistical analysis – The used tests refer to a normal distribution of the data; however, it is not stated based on which test this was observed; please clarify this. 

Results

Line 168. Regarding female predominance, I would mention that this is generally observed in neurotological disorders.

Line 171. Treating bilateral cases would be interesting to detail, which can be challenging in everyday practice.

Table 1. First, the essential information in this table should have been discussed in the text. I.e., the statistically insignificant differences in gender, age and canal involvement between the two groups must be stated. This refers to the fact that the two groups are statistically comparable. Furthermore, the significance of BMI related to this content must be clarified. Otherwise, there is no rationale to include this parameter. In addition, the duration of symptoms and previously used CRMs should be included in this table; the latter is especially significant considering a possible multicanal involvement. Further minor necessary corrections: p-value instead of ‘p’, include each abbreviation along with the statistical testing used and significance level in the table caption. 

Lines 180-181. It would be interesting to specify these insignificant differences, i.e., the comorbidities and possible risk factors were statistically insignificant between the two groups. Furthermore, the group ‘history of BPPV’ must be clarified. I.e., do the authors mean a previous BPPV in the case history or a not-treated BPPV related to the current vertigo episode? Furthermore, each abbreviation, along with the statistical testing used and significance level, must be included in the table caption. 

Figure 2. The p-value showing a statistically insignificant difference should be included in this figure to interpret the results more easily. Consequently, statistical testing and significance level must be included in the figure caption. The same corrections are applicable in the case of Figure 2.

Lines 202-203. It is unsure to me how correlations between continues variables were observed without mentioning a correlation test. Furthermore, in addition to the p-value, a correlation coefficient must also be included.

Table 4. First, the term CRM should consequently be used; I do not recommend writing ‘manual treatment’ as it might be misleading. Furthermore, each abbreviation, along with the statistical testing used and significance level, must be included in the table caption. ‘p’ should be corrected to ‘p-value’. Regarding the DHI, it must be specified that the authors included the mean total DHI scores here. 

Figure 5. I recommend including the p-value in this figure and explaining the abbreviations in the figure caption.

Discussion

‘don’t have’ – Please correct this to do not have; this form should not be used in scientific English writing. There are also other examples in the discussion.

Lines 255-257. This sentence may refer to a concern, i.e., in the methods, it was not stated that all patients received a manual CRM during the first visit, and they were later divided into two groups. A previously used treatment can significantly impact the generalisation of the results of this investigation; therefore, this must be clarified. 

Generally, when discussing previous results, a more comprehensive presentation would be beneficial, i.e., a significant difference was observed between two groups in which aspect, etc.

From my point of view, to highlight the importance of accurate BPPV diagnosis in acute presentation, I recommend mentioning that the objective neurotological testing results in BPPV are still controversial. This emphasises the importance of the earliest possible diagnosis based on conventional CRMs or TRV. Regarding this, I recommend including the following reference article: 

doi: 10.1016/j.joto.2021.11.001.

Conclusions

The importance of avoiding later BPPV recurrences would be beneficial to mention.

‘…’ I do not recommend to use this in scientific writing.

I am looking forward to receiving the revised version of the manuscript, which includes a point-by-point response to each review comment with all required changes accurately made. This is necessary for deciding whether this manuscript can be considered for publication. 

Comments on the Quality of English Language

Minor editing is necessary.

Reviewer 2 Report

Comments and Suggestions for Authors

In this manuscript, Chaure-Cordero et al. report the findings of a prospective case study that investigated the effectiveness of mechanical rotational chairs (TRV) in the treatment of BPPV with that of conventional repositioning maneuvers. The study had 63 patients that were randomly assigned to one of the treatment groups after initial diagnosis and conventional treatment in the emergency room. The authors report that there were no statistically significant differences between the groups expect for a correlation with age, older patients requiring less maneuvers in the TRV group. The methodology and results have been present well. The conclusion is supported by the presented results. 

Minor comments:

1) Line 196-197 states 16 + 16 = 32 patients with history of BPPV. In table 2 it is stated as 16 + 19 = 35. Either the sentence or table need to be corrected.

2) For the patients with history of BPPV, where they treated by CRM or TRV previously? How long ago was the previous episode?

Round 2

Reviewer 1 Report

Comments and Suggestions for Authors

Thank you for the revised version of the manuscript. The authors have made efforts to improve the quality of this manuscript, which now reads better. Only a few minor corrections remain, and this article can be considered for publication. 

Line 63. 'It's' must be corrected to 'it is'. Change "It's" to "It is" on line 63. Also, correct other instances of contractions like 'don't' or 'didn't'.

Rephrase the sentence on lines 67-68 to convey the intended meaning accurately.

Use the correct spelling for Ménière's disease on line 139.

Figure 2. In addition to the significance level, the name of the statistical test should also be included. 

Lines 272-273. It would be beneficial to state that this is a significant correlation; however, the correlation coefficient refers to a weak correlation. 

Table 4. The statistical test name and the significance level are missing.

Figure 5. The statistical test name is missing. 

Lines 305-306. It would be beneficial to state that this is a significant and moderate correlation. 

Line 337. Regarding the study design, a limitation must be stated. For instance, it can significantly impact the results that each patient received a conventional canalith repositioning manoeuvre during the first visit. As a further study plan, it would be beneficial to consider creating two groups, one receiving conventional and one TRV treatment during the first visit. This could significantly increase the general usability of the results, providing a more comprehensive understanding of the study findings. 

Comments on the Quality of English Language

Minor editing is necessary. I have included some examples above. 
